# The effect of social support on academic performance among adolescents: The chain mediating roles of self-efficacy and learning engagement

Xiangping Zhang[1,2], Wensheng Qian [3]*

1 Faculty of Education, Qufu Normal University, Qufu, Shandong, China, 2 Faculty of Tourism and Art, Xiangyang Polytechnic, Xiangyang, China, 3 Faculty of Architecture and Engineering, Xiangyang Polytechnic, Xiangyang, China

* qianwensheng2022@163.com

**Data Availability Statement:** All relevant data are within the manuscript and its Supporting Information files.

## Abstract

### Purpose

While the impact of social support on academic performance is acknowledged, the specific mechanisms by which social support affects academic performance, particularly through self-efficacy and learning engagement, remain poorly understood. This study aims to examine the correlation between social support and academic achievement among Chinese middle school students, framed within the Social Cognitive Theory. It also seeks to explore the mediating roles of self-efficacy and learning engagement in this relationship.

### Method

Data was collected from 265 individuals (mean age = 13.47 years, SD = 0.5) in four middle schools in Shandong Province, China in June 2023, using the simple random sample technique. Participants completed the questionnaires independently, and the data was analyzed using the structural equation model (SEM) in AMOS 24.0 and SPSS 24.0.

### Results

Social support and academic performance have a direct and significant relationship with the SCT among middle school students. In addition, social support indirectly and positively affects academic performance through self-efficacy and learning engagement. The results also highlight self-efficacy as a key factor linking social support with academic performance.

### Practical implications

This study offers valuable insights into the role of social support in Chinese middle school students' academic achievement, particularly by examining the impact of self-efficacy and learning engagement. These valuable findings may guide policymakers in creating a supportive educational environment both inside and outside the classroom, enhancing adolescents' self-confidence and engagement in learning.

**Funding:** The author(s) received no specific funding for this work.

**Competing interests:** The authors have declared that no competing interests exist.

## Originality

This study contributes to the theoretical understanding of social support by investigating the mechanisms through which it impacts academic achievement. It clarifies the complex interactions among social support, self-efficacy, learning engagement, and academic achievement, with particular emphasis on the mediating roles of self-efficacy and learning engagement within the Chinese context.

## Introduction

Within China's compulsory education system, middle school serves as a crucial intermediary, bridging different levels of education. At this stage, academic performance is a significant indicator of students' information acquisition and their potential for future studies. It is used to evaluate student progress, learning, and talent selection [1, 2]. In China, academic success is commonly assessed through exam scores in subjects such as Chinese, Math, and English [3, 4]. Nevertheless, this assessment approach presents challenges for middle school students in developing study routines, overcoming academic difficulties, and managing time and anxiety [5]. Learning is predominantly a cognitive endeavor that involves the acquisition of knowledge through various social activities [6]. Chinese students, however, have limited opportunities for social engagement, primarily confined to interactions within the classroom, on campus, and within their families. Social support refers to the resources obtained through social interactions, which reflect the degree of connection between an individual and their community. It serves as a defensive shield against negative emotions and stress [7, 8], providing a sense of being valued and supported by others when needed [9]. The concept has three dimensions: subjective support, objective support, and support usage [10]. Adolescents rely on certain social support to effectively manage stress, anxiety, and other psychological challenges that hinder their academic progress [11–13]. Prior studies have examined how social support affects adolescents' learning behaviors, attitudes, motivation, and ultimately, academic performance [14, 15]. Academic performance generally refers to the grades, academic achievements, abilities, and learning outcomes demonstrated by students, assessed through various criteria [16, 17]. It has significant implications for different aspects of a student's educational journey and prospects [18, 19]. In this study, academic performance specifically refers to students' test scores in Chinese, English and Math.

Although previous researchers have examined the impact of social support on academic performance, less attention has been paid to the roles of self-efficacy and learning engagement. These two psychological constructs are closely related to academic performance [20]. Self-efficacy acts a crucial role in motivating individuals to achieve their goals, encouraging them to take risks, and reaching their academic outcomes [21, 22]. According to Liem et al. [23], students with higher levels of self-efficacy are more likely to participate actively in social activities and fully engage in deep learning. Moreover, students who possess a strong belief in their own academic capabilities are more inclined to feel motivated, persevere in the face of difficulties, and establish ambitious academic goals [24]. Learning engagement refers to the active participation of students in the educational process, which positively impacts on academic performance. Engaged students are typically more motivated, committed, and willing to invest the effort needed to participate in discussions, solve problems, and achieve academic success [25, 26]. These findings highlight the importance of considering self-efficacy and learning engagement when studying academic achievement. However, little attention has been paid to the

influence of social support on the academic performance of Chinese middle school students. Furthermore, this study identifies a gap in understanding the precise mechanisms by which social support affects academic performance through self-efficacy and learning engagement, as outlined in relevant theoretical frameworks.

The Social Cognitive Theory (SCT), proposed by Bandura in 1986, serves as the theoretical framework for developing a chain mediation model in this study. According to this theory, human behavior is influenced by three variables: personal factors (e.g., self-efficacy), behavioral factors (e.g., learning engagement and achievement), and environmental factors (e.g., social support). Namely, individual behavior is shaped and moderated by the interaction of these personal, behavioral, and environmental elements [27]. Previous research has applied SCT to explore how personal cognitive and environmental factors influence academic performance among adolescents [28, 29]. However, less attention has been given to the interplay between social support, self-efficacy, learning engagement, and academic performance within the SCT framework. This study, therefore, aims to provide a comprehensive understanding of the combined influence of these four factors within the context of SCT.

This study adopts a broader approach by examining the interrelationships and mediating effects of social support, self-efficacy, learning engagement, and academic performance. Unlike previous research, which often focused on the impact of individual factors on academic performance, this study explores the combined effects of these variables. Its significance lies in bridging existing research gaps and enhancing our understanding of the factors that contribute to students' academic success.

## Literature review

### Social support and academic performance

Relevant research has demonstrated that social support is a significant predictor of academic performance. Studies have found that social support can substantially boost individuals' self-confidence [30], and motivation [31]. Also, researchers have shown that parental involvement and encouragement can increase students' focus and motivation, leading to improved academic outcomes [32, 33]. In addition, a supportive family environment provides stability and emotional support, which helps students achieve higher academic grades [34]. Social support from peers is another crucial factor in influencing academic performance [35, 36]. For instance, Wentzel [37] noted that interactions with peers who exhibit positive learning attitudes and behaviors can stimulate students' motivation and enhance their academic performance. Importantly, it has been revealed that social support from educators has a significant positive impact on academic performance [38]. Research suggests that individualized tutoring and additional assistance from teachers can address students' specific learning needs and provide effective guidance, thereby promoting academic progress [39]. In sum, these findings underscore the critical role of social support in academic performance, highlighting that adolescents who receive support from parents, peers, and teachers are more likely to succeed in their academic pursuits. Based on this, we propose the following hypothesis:

H1: Social support is positively correlated with academic performance.

### Self-efficacy as a mediator

Social support has been shown to play a crucial role in shaping self-efficacy beliefs. Research indicates that support from family, peers, and teachers can positively influence students' self-efficacy [39–42]. For example, students who receive encouragement, guidance, and positive feedback from family members are more likely to develop a strong sense of self-efficacy in

their academic abilities [43, 44]. Similarly, interactions with supportive peers and teachers who provide assistance and express confidence in students' capabilities can further strengthen their self-efficacy beliefs [45, 46]. Moreover, self-efficacy is a significant factor in students' academic achievement [47]. Students with higher self-efficacy are more likely to set ambitious goals, study diligently, and persist through unforeseen challenges. They are also more inclined to use effective learning strategies, seek help when needed, and overcome obstacles [48]. Conversely, students with low self-efficacy may doubt their abilities, experience increased anxiety, and achieve lower academic results [49]. Therefore, understanding the relationship between self-efficacy and academic achievement is critical for examining students' academic success. Bai et al. [50] have suggested that self-efficacy mediates the relationship between social support and English learning performance among secondary students in Hong Kong. They argue that social support influences students' self-efficacy beliefs, which, in turn, affect their academic performance. Based on the literature reviewed, the following hypothesis is proposed:

H2: Self-efficacy mediates the relationship between social support and academic performance.

## Learning engagement as a mediator

Learning engagement is significantly influenced by social support. Mishra has put forth the notion that students' learning engagement can be potentially enhanced by social support in the learning process [51]. For example, both academic and emotional support from peers can greatly contribute to students' learning engagement [52]. Wong et al. [53] have asserted that teacher support can effectively stimulate students' interest, enjoyment, dedication, investment of time, effort, emotions, and learning strategies. Additionally, Shao and Kang [41] have suggested that parental support can influence adolescents' level of involvement in their learning endeavors.

Meanwhile, learning engagement is considered to be an important factor that affects students' academic performance. High levels of learning engagement allow students to dedicate more time to learning activities, leading to better academic outcomes [54–56]. Classroom engagement, in particular, has been shown to have a significant impact on academic achievement [39]. Therefore, this study proposes that learning engagement may serve as a mediator between social support and academic achievement.

Self-efficacy is also believed to play a key role in increasing student involvement in learning activities [57, 58]. Students with high levels of self-efficacy are more likely to establish ambitious objectives and actively participate in learning activities [59, 60]. According to the SCT, the surrounding environment can impact personal cognition (e.g. self-efficacy) and behavior (e.g. learning engagement, academic performance). When social support is strengthened, adolescents may improve their self-efficacy, leading to better engagement in learning and, ultimately, better academic outcomes. Thus, we hypothesize that social support may influence academic performance through the sequential mediation of self-efficacy and learning engagement. The following hypotheses are derived:

H3: Self-efficacy mediates the relationship between social support and adolescents' academic performance.

H4: Self-efficacy and learning engagement play a chain mediating role in the association between social support and adolescents' academic achievement.

Under the guidance of the above hypotheses and Social Cognitive Theory (SCT), a theoretical model was developed to investigate the relationship between social support and adolescents' academic performance, as well as the mediating roles of self-efficacy and learning engagement (Fig 1).

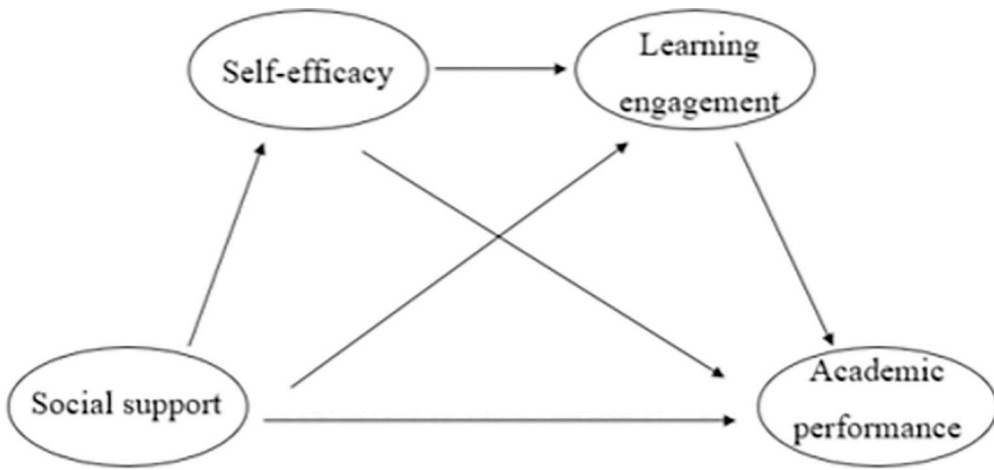

**Fig 1. The theoretical model.**

## Materials and methods

### Sampling and data collection

Before commencing the research, ethical considerations were prioritized. The Institutional Review Board (IRB) at Qufu Normal University carefully reviewed and approved the research procedures, ensuring that the rights and welfare of all participants were upheld. Informed consent was obtained from both students and their parents prior to participation in the survey. Additionally, consent was secured from the headmasters of the sampled schools.

The study employed a simple random sampling method to recruit Chinese middle school students who voluntarily participated in the questionnaire collection. Data was collected from June 1 to June 30, 2023. First, a structured questionnaire was administered to students during regular school hours in a classroom setting. The researchers provided clear instructions and addressed any questions or concerns from the participants. Students were encouraged to respond honestly, with the assurance of confidentiality regarding their responses. Second, the study included the collection of students' academic performance data. Academic performance was measured using scores from the final exams in Chinese, Math, and English subjects. To ensure comparability and facilitate analysis across different subjects, the overall scores, ranging from 0 to 120, were standardized. These standardized scores were used as observed variables to measure academic achievement in the study.

A proper sample size, at least ten times the total number of observed variables, was required based on the recommendations for Structural Equation Modeling (SEM) [61]. We distributed 300 survey forms to participants aged 13–15 from four middle schools in Shandong Province, China. A total of 265 completed survey forms were received (92.6% response rate), with 14 surveys (4.7%) rejected due to incomplete records and missing answers. The selection of these four middle schools was based on their willingness to participate. Additionally, these schools are representative of diverse socioeconomic backgrounds, including a mix of urban and rural locations, varied student populations, and diverse academic performance levels. This ensured a balanced representation of both genders and different grade levels within the specified age range.

### Sample characteristics

The sample consisted of 265 participants from four middle schools in eastern China. The average age of the participants was 13.71 years (SD = 0.5, range = 13–14 years). As shown in

Table 2, there were 128 female students (48.3%) and 137 male students (51.7%). The sample included 93 first-year students (35.1%), 91 second-year students (34.3%), and 81 third-year students (30.6%). Regarding residence, 137 families (51.7%) lived in urban areas, while 128 families (48.3%) lived in rural areas. Concerning parental education, 159 fathers (60%) and 192 mothers (72.5%) had completed junior high school or less. Seventy-seven fathers (29.1%) and 51 mothers (19.2%) had received education at the high school or technical school level. Additionally, 17 fathers (6.4%) and 12 mothers (4.5%) had received education at the college level, while 12 fathers (4.5%) and 10 mothers (3.8%) had received education at the undergraduate level or above. In terms of monthly income, 52 families (19.6%) had an income below 3,000 yuan, 124 families (46.8%) had an income between 3,000 and 5,000 yuan, 68 families (25.7%) had an income between 5,000 and 10,000 yuan, and 21 families (7.9%) had an income above 10,000 yuan. Overall, the sample in this study is representative of the target population, as it closely matches the demographic features and distribution observed in the region.

## Questionnaire design

The questionnaire employed in the study was based on well-established tools that have been demonstrated to be reliable and valid. It consisted of two primary sections. The first section gathered demographic data such as gender, grade level, place of residence, and parental education level. This information was crucial for describing the sample and providing context for the analysis. The second section contained 15 items, carefully selected from validated and pre-existing scales measuring social support, self-efficacy, and learning engagement. Each item was chosen based on its relevance to the study's objectives and its proven utility in previous research. Five items for assessing social support were derived from Ye and Dai's Social Support Scale [10], chosen for their relevance to assessing perceived levels of social support among participants. Three items measuring self-efficacy were obtained from Schwarzer's study [62], selected for their ability to capture participants' beliefs in their academic abilities. Five items evaluating learning engagement were selected from Fang et al.'s Learning Engagement Scale [63], with adjustments made according to the Utrecht Work Engagement Scale-Student [64]. These items were chosen to assess participants' levels of involvement, interest, and emotional connection to the learning process. All 15 items were scored on a 5-point Likert scale ranging from 1 (strongly disagree) to 5 (strongly agree). This scale allowed participants to rate their agreement with each statement. Specific details on the measurement items can be found in Table 1.

## Statistical analysis

Data were analyzed using Amos 24.0 and SPSS 24.0. First, the Harman single-factor test was conducted to assess the potential for common method bias. Descriptive analysis was then employed to appropriately reflect the characteristics of the sample. Subsequently, structural equation modeling (SEM) was used to evaluate both the measurement and structural models. The measurement model was validated through confirmatory factor analysis, while the structural model was examined using goodness-of-fit indices and path coefficients. Finally, the significance of mediating effects was assessed using the bootstrapping method.

## Results

### Common method variance

A Harman single-factor test was conducted to assess the potential impact of common method variance. This analysis involved performing an exploratory factor analysis on all items from

Table 1. Potential variables and measurement items.

| Potential variable | Code | Measurement items | References |
|---|---|---|---|
| Social support (SS) | SS1 | Most of my classmates care about me. | Ye and Dai (2008) |
| | SS 2 | I can often count on the care and support of my classmates, friends, parents, and others. | |
| | SS 3 | I am surrounded by people who are close and can give me support and help. | |
| | SS 4 | Teachers, classmates, friends, family, etc. are there for me when I have problems. | |
| | SS5 | I often get emotional help and support from my classmates and friends. | |
| Self-efficacy (SE) | SE1 | If I do my best, I can always solve problems. | Fang et al. (2008) |
| | SE2 | It is simple for me to pursue my dreams and achieve my objectives. | |
| | SE3 | I can face difficulties calmly because I am confident in my ability to deal with them. | |
| | SE4 | When there is a problem, I usually come up with solutions. | |
| | SE5 | Whatever happens to me, I can handle it. | |
| Learning engagement (LE) | LE1 | As soon as I wake up in the morning, I'm happy to study. | Schaufeli (2002) |
| | LE2 | I find learning purposeful and rewarding. | |
| | LE3 | I am passionate about my learning. | |
| | LE4 | I am proud of my learning. | |
| | LE5 | I find learning challenging. | |

the three scales using an unrotated principal component analysis approach. The results indicated that three components had eigenvalues greater than one, with the first factor accounting for 23.704% of the total variance. This percentage falls short of the critical threshold of 50% [65]. As a result, we may infer that there is no significant common method bias in this study.

## Measurement model

The measurement model provides a framework for analyzing the links between the observed indicators and the underlying components [61]. In this analysis, it is essential to evaluate both reliability and validity. Reliability is commonly assessed using Cronbach's alpha coefficient, with values ranging from 0.80 to 0.89 considered satisfactory. Convergent validity is measured using indicators such as average variance extracted (AVE), standardized component loadings, and composite reliability (CR), with a threshold of 0.50 or higher deemed appropriate [66]. Discriminant validity is established by examining the correlations among various constructs and comparing these correlations to the square root of the AVE for each construct. If the correlations between constructs are less than the square root of the AVE for each construct, this indicates that the assessment items measure distinct constructs and demonstrate discriminant validity [67].

The reliability and convergent validity analysis results are presented in **Table 2**. Cronbach's alpha coefficients ranged from 0.876 to 0.892, indicating that the measurement model is highly reliable. Additionally, standardized factor loadings varied from 0.689 to 0.880, reflecting strong convergent validity. The composite reliability (CR) and average variance extracted (AVE) values ranged from 0.880 to 0.895 and from 0.595 to 0.633, respectively, indicating good convergent validity. Table 3 shows that the AVE values for each variable exceeded the squared correlation coefficients in the corresponding rows and columns. This finding confirms that the measurement items in this study exhibit robust reliability and convergent validity. To validate discriminant validity, this study employed the Heterotrait-Monotrait (HTMT) criterion. According to Kline [68], and Henseler et al. [69], an acceptable HTMT value should remain below 0.85. **Table 4** indicates that discriminant validity was achieved. Table 5 shows that the path coefficient value $R^2$ shows that only 14.1% of the total variance was explained by social support, learning engagement and academic performance. Furthermore, $R^2$ shows that 25.9%

**Table 2. Reliability and validity.**

| Latent variable | Item | UC | SE | Z-value | P-value | SC | Cronbach's a | CR | AVE |
|---|---|---|---|---|---|---|---|---|---|
| Social support (SS) | SS 1 | 1.000 | | | | 0.767 | | | |
| | SS 2 | 0.890 | 0.079 | 11.272 | *** | 0.689 | | | |
| | SS 3 | 0.941 | 0.079 | 11.885 | *** | 0.722 | 0.876 | 0.880 | 0.596 |
| | SS4 | 1.087 | 0.083 | 13.093 | *** | 0.787 | | | |
| | SS 5 | 1.115 | 0.076 | 14.709 | *** | 0.880 | | | |
| Self-efficacy (SE) | SE 1 | 1.000 | | | | 0.755 | | | |
| | SE 2 | 1.072 | 0.082 | 13.038 | *** | 0.791 | | | |
| | SE 3 | 1.160 | 0.083 | 13.945 | *** | 0.842 | 0.892 | 0.895 | 0.633 |
| | SE4 | 1.152 | 0.080 | 14.378 | *** | 0.867 | | | |
| | SE5 | 1.047 | 0.090 | 11.594 | *** | 0.712 | | | |
| Learning engagement (LE) | LE1 | 1.000 | | | | 0.734 | | | |
| | LE2 | 0.905 | 0.077 | 11.809 | *** | 0.756 | | | |
| | LE3 | 1.013 | 0.086 | 11.843 | *** | 0.758 | 0.878 | 0.880 | 0.595 |
| | LE4 | 1.068 | 0.081 | 13.259 | *** | 0.854 | | | |
| | LE5 | 0.960 | 0.082 | 11.701 | *** | 0.749 | | | |

UC = Unstandardized Coefficients; SE = standard error; SC = standardized coefficients

***p < 0.001.

and 47.4% of the total variance in learning engagement and academic performance were explained by social support.

## Structural model

The study utilized the maximum likelihood estimation method in AMOS 24.0 software to examine the model. The fit indices for the data and model are as follows: $\chi2 = 194.930$, df = 129, $\chi2/df = 1.511$, GFI = 0.927, AGFI = 0.903, IFI = 0.977, TLI = 0.972, CFI = 0.976, RFI = 0.922, NFI = 0.934, SRMR = 0.0434, and RMSEA = 0.044. All the values met the recommended thresholds [61], indicating a good fit for the structural model. Furthermore, sensitivity analysis revealed an effect size of 0.39, meeting Cohen's threshold for a strong statistical test with a sample size of 265 [70].

**Fig 2** shows the structural model's standardized parameter estimates, which include explanatory variance and path coefficients. The study found that the social support construct explains 14% of the variance in self-efficacy, with a standardized regression value of 0.441 *(P < 0.001)*. The social support and self-efficacy constructs together explain 26% of the variance in the learning engagement construct, with standardized regression coefficients of 0.424 *(P < 0.001)* and 0.155 *(P < 0.01, respectively)*. The social support, self-efficacy, and learning engagement variables had a significant impact on academic achievement, accounting for 47% of the variation (standardized regression coefficients of 0.298 *(P < 0.001)*, 0.292 *(P < 0.001)*, and 0.218

**Table 3. Discriminate validity examination.**

| Potential variable | Social support | Self-efficacy | Learning engagement |
|---|---|---|---|
| Social support | **0.772** | | |
| Self-efficacy | 0.374*** | **0.795** | |
| Learning engagement | 0.481*** | 0.333*** | **0.771** |

Note: The diagonal shows the square root of the AVE of four latent constructs, whereas the diagonal below shows the correlation coefficient.

**Table 4. Heterotrait-Monotrait ration (HTMT).**

| Potential variable | Social support | Self-efficacy | Learning engagement |
|---|---|---|---|
| Social support | | | |
| Self-efficacy | 0.403 | | |
| Learning engagement | 0.508 | 0.326 | |

($P < 0.001$). These empirical findings provide substantial support for the proposed structural model.

### Analyzing the mediation of social support on academic performance

This study used the Bootstrap technique to look into the function of self-efficacy and learning engagement as mediators in the connection between social support and academic performance. Following Hayes' recommendation [71], a bootstrap sample size of 2000 was set, with a confidence level of 95%. A mediating effect is considered statistically significant when the Bias-Corrected and Percentile methods' confidence intervals, at a 95% confidence level, do not include zero [72]. Data analysis was performed using Amos 24.0 software. In this analysis, academic performance was treated as the dependent variable, while social support was considered the independent variable. Additionally, self-efficacy and learning engagement were regarded as mediating variables. **Table 6** displays the findings of the mediation study, which identified self-efficacy and learning engagement as mediators of the association between social support and academic performance.

At a 95% confidence level, the Bias-Corrected and Percentile methods' confidence intervals do not include zero, indicating that social support has a significant total effect (bias-corrected CI [0.394, 0.673], percentile CI [0.393, 0.672], $P < 0.01$) and a significant direct effect (bias-corrected CI [0.152, 0.440], percentile CI [0.153, 0.441], $P < 0.01$) on academic performance. Furthermore, the analysis revealed significant indirect effects in three pathways. The indirect effect of social support→self-efficacy→learning engagement→academic performance was 0.015 (95% bias-corrected CI [0.005, 0.038], percentile CI [0.003, 0.032], $P<0.01$). The indirect effect of social support→self-efficacy→academic performance was 0.129 (95% bias-corrected CI [0.072, 0.205], percentile CI [0.068, 0.199], $P<0.01$). Lastly, the pathway of social support→learning engagement→academic performance had an indirect effect of 0.092 (95% bias-corrected CI [0.042, 0.171], percentile CI [0.036, 0.158], $P<0.01$). The results indicate that the three mediating effects were all statistically significant, providing support for H2, H3, and H4. Moreover, the direct effect accounted for 55.8% of the total effect, while the three indirect effects collectively accounted for 44.2% of the total effect. Notably, among the three indirect effects, the pathway "social support → self-efficacy → academic performance" exhibited the strongest effect.

### Discussion

According to SCT, the present study looked into the link between social support, self-efficacy, learning engagement, and academic performance among adolescents. To understand the

**Table 5. Coefficient of determination R2 variance inflation factor.**

| Constructs | $R^2$ | VIF |
|---|---|---|
| Self-efficacy | 0.141 | 1.353 |
| Learning engagement | 0.259 | 1.249 |
| Academic performance | 0.474 | 1.537 |

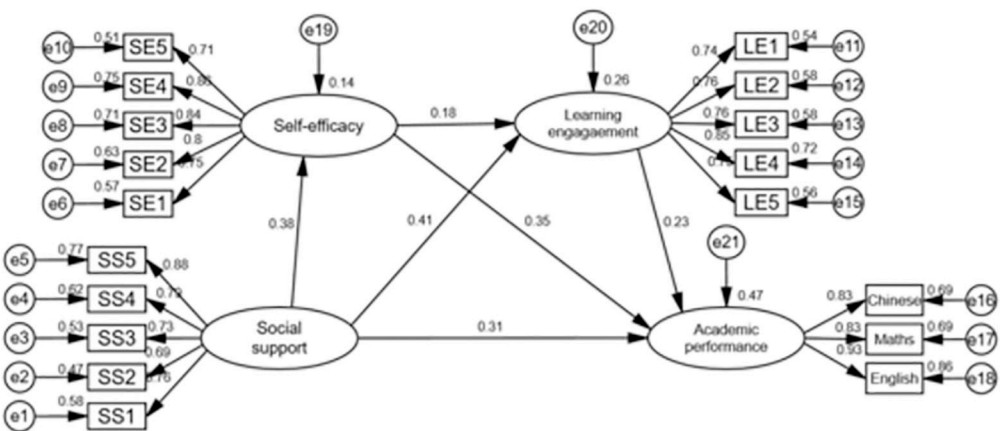

**Fig 2. The structure modeling diagram.**

mechanism through which social support affects academic performance, the study sought to investigate the potential mediating effects of self-efficacy and learning engagement in the relationship between social support and academic performance in the exam-oriented education setting. The study's findings are presented below.

## Direct effects analysis

Our results indicate that social support has a positive and direct impact on adolescents' academic performance. This finding is consistent with previous studies conducted by Song et al. and Jacobson and Burdsal [31, 73]. Besides, Gallardo et al. [74] hypothesized a positive correlation between social support and academic performance among middle school students. Escalante et al. [75] further reinforced this hypothesis by demonstrating that academic

**Table 6. Total, direct, and indirect effects of the theoretical model.**

| Path relationship | | Point estimate | Product of coefficient | | Bootstrapping | | | |
|---|---|---|---|---|---|---|---|---|
| | | | | | Bias-corrected 95% CI | | Percentile 95% CI | |
| | | | SE | Z-value | Lower | upper | lower | upper |
| Test of indirect, direct and total effects | | | | | | | | |
| DistalIE | SS→SE→LE→AP | 0.015 | 0.008 | 1.875 | 0.005 | 0.038 | 0.003 | 0.032 |
| LMIE | SS→SE→AP | 0.129 | 0.034 | 3.794 | 0.072 | 0.205 | 0.068 | 0.199 |
| LEIE | SS→LE→AP | 0.092 | 0.031 | 2.968 | 0.042 | 0.171 | 0.036 | 0.158 |
| TIE | Total indirect effect | 0.236 | 0.051 | 4.627 | 0.154 | 0.358 | 0.142 | 0.345 |
| DE | SS→AP | 0.298 | 0.075 | 3.973 | 0.152 | 0.440 | 0.153 | 0.441 |
| TE | total effect | 0.534 | 0.070 | 7.629 | 0.394 | 0.673 | 0.393 | 0.672 |
| Percentage of indirect effects | | | | | | | | |
| P1 | DistalIE/TIE | 0.063 | 0.028 | 2.250 | 0.021 | 0.134 | 0.014 | 0.124 |
| P2 | SEIE/TIE | 0.546 | 0.105 | 5.200 | 0.359 | 0.763 | 0.360 | 0.766 |
| P3 | LEIE/TIE | 0.391 | 0.098 | 3.990 | 0.188 | 0.577 | 0.188 | 0.577 |
| P4 | TIE/TE | 0.442 | 0.099 | 4.465 | 0.288 | 0.685 | 0.272 | 0.656 |
| P5 | DE/TE | 0.558 | 0.099 | 5.636 | 0.315 | 0.712 | 0.344 | 0.728 |

Note: SS = Social Support, SE = Self-efficacy, LE = Learning Engagement, AP = academic Performance, IE = Indirect effect, TIE = Total Indirect Effect, DE = Direct Effect, TE = Total Effect, DIE = Distal Indirect Effect

performance is influenced by school climate, with social support being the dominant factor. This study confirms these findings by emphasizing the positive role of social support in adolescents' academic performance. One possible explanation for this effect is that Chinese adolescents, facing intense competition and learning pressures, may better manage the evaluation of threatening situations and boost their self-confidence when they receive timely social support. This support can encourage them to persist in overcoming learning challenges [76]. Another explanation is that adolescents who perceive social support may experience increased feelings of security and hope in their learning environment, which in turn motivates them to engage more actively in the learning process, thereby contributing to improved academic performance [77]. This study further validates social support as a predictive factor for academic performance.

### Indirect effects analysis

The study found that self-efficacy partially mediates the relationship between social support and academic performance among Chinese adolescents. In accordance with Social Cognitive Theory (SCT), this finding underscores the crucial role of self-efficacy as a mediator in the pathway from social support to academic performance, consistent with prior research [78]. Under the intense pressure of examinations, Chinese middle school students who receive greater social support from peers, teachers, and parents are likely to develop the self-efficacy needed to succeed in tests and to estimate the effort required to reach their academic goals [79]. Moreover, improved relationships with parents, teachers, and classmates can reduce student stress, enhance problem-solving skills, and promote positive, self-directed behavioral patterns [80]. Adolescents who experience these benefits in their learning behaviors show higher levels of self-efficacy, which subsequently boosts their academic success. By highlighting its significant mediating role, the findings of this study further validate the importance of self-efficacy in achieving academic excellence [38]. This evidence reinforces the substantial mediating effect that self-efficacy has on the relationship between social support and academic achievement among adolescents.

The results of the study demonstrated that learning engagement also partially mediated the association between social support and academic performance among adolescents. This suggests that high levels of learning engagement may clarify why middle school students cultivate positive social factors, such as interaction with peers, teachers, and parents, to enhance their academic performance. Students who receive positive social support are more likely to engage actively in their studies, as evidenced by their eagerness to complete tasks, participation in class, and proactive pursuit of new learning opportunities, ultimately leading to improved academic outcomes [81]. Moreover, establishing connections with positive social factors fosters a supportive environment, increases adolescents' engagement in learning, and enhances academic performance [82, 83]. These findings align with previous studies [52, 84, 85], which posit that learning engagement is a key factor linking social support and adolescents' academic achievement.

### Chain mediator effect

One of the most unexpected findings of the study was the revelation that learning engagement and self-efficacy functioned as a chain mediator in the relationship between social support and academic performance. This result aligns with Bandura's Social Cognitive Theory (SCT), illustrating how supportive relationships and nurturing interactions with peers, parents, and educators create a positive environment conducive to the development of adolescents' self-efficacy [78]. This self-efficacy, in turn, influences their level of engagement in learning, which leads to

better academic performance. Furthermore, the study revealed that adolescents' self-efficacy had a lesser impact on their level of learning engagement ($\beta = 0.17$, $P < 0.001$) compared to the influence of social support ($\beta = 0.41$, $P < 0.001$). This suggests that the primary source of learning engagement among adolescents is social support. Interaction with peers, teachers, parents, and others provides a supportive learning environment that enhances students' self-efficacy in participating in educational activities [43].

## Conclusion

The study aimed to examine the relationship between social support and academic performance within the context of China's test-based culture. The current findings indicate that adolescents' academic performance is positively and directly influenced by social support. Also, learning engagement and self-efficacy function as chain mediators in the association between social support and academic performance. The relationship and effects of these various variables are further elucidated through the lens of Social Cognitive Theory (SCT). As demonstrated by previous research, this article is distinctive and offers new insights into the roles of social support, self-efficacy, learning engagement, and academic performance within the particular academic climate of China.

### Theoretical implication

This study contributes to the existing theoretical knowledge by underscoring the influence of social support on the academic performance of middle school students within the framework of Social Cognitive Theory. It supports the notion that social support plays a significant role in academic achievement by elucidating the complex interactions among social support, self-efficacy, learning engagement, and academic performance. Furthermore, this research builds on previous empirical studies that have established a link between social support and academic performance. By confirming these findings within the context of the Chinese compulsory education system and emphasizing the mediating roles of self-efficacy and learning engagement, this study enhances the theoretical understanding of the relationship between social support and academic achievement among middle school students in China.

### Practical implication

Regarding the practical implications, it is essential for educational practitioners to understand how to enhance students' academic achievement by considering social factors such as the roles of teachers, parents, and peers. To strengthen social support, teachers should create an inclusive and cohesive classroom environment that fosters respect, understanding, and collaboration among students. This can be accomplished through initiatives like peer mentorship programs and collaborative learning activities. Parents also play a vital role in establishing a conducive learning environment at home. They can do this by promoting a focused atmosphere, designating a dedicated study area, and minimizing external distractions. To enhance self-efficacy, it is important for both teachers and parents to encourage students to participate in problem-solving activities that relate to real-life situations. Additionally, they should motivate adolescents to embrace challenges and seek solutions, thereby helping them develop confidence in their abilities [86]. Furthermore, it is essential for educators and guardians to provide timely and constructive feedback that allows students to monitor their learning progress and adjust their approaches accordingly. This kind of feedback can significantly enhance students' self-efficacy and belief in their own abilities. In terms of learning engagement, it is important for teachers, parents, and other social variables to collaborate in order to develop a comprehensive understanding of adolescents' needs. By employing effective strategies and

techniques, they can foster greater involvement in learning through meaningful and practical activities. This coordinated effort will not only engage students more deeply but also support their overall academic development.

## Limitations and further research

There are several limitations to acknowledge in this study. First, the use of a cross-sectional design restricts the ability to establish causal relationships between the examined factors. Therefore, longitudinal research is required to bridge a definitive link between social support and academic performance over time. Second, the study was conducted within China's test-oriented learning environment, which may limit the applicability of the findings to other educational contexts. To strengthen the study's external validity, subsequent research should be undertaken in multiple countries and diverse educational settings. Other relevant factors, such as academic flow, academic resilience, and learning motivation, were not considered in this study. Future investigations should incorporate these elements into a more comprehensive theoretical framework to provide an insightful view of the dynamics involved.

## Supporting information

**S1 Data.**
(XLSX)

## Author Contributions

**Investigation:** Wensheng Qian.

**Methodology:** Wensheng Qian.

**Project administration:** Wensheng Qian.

**Supervision:** Xiangping Zhang.

**Writing – original draft:** Xiangping Zhang.

**Writing – review & editing:** Xiangping Zhang, Wensheng Qian.

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
