## [Decision Letter · Decision Letter 0]

4 Jun 2024

PONE-D-24-10692The effect of social support on academic performance among adolescents: The chain mediating roles of self-efficacy and learning engagementPLOS ONE

Dear Dr. Qian,

Thank you for submitting your manuscript to PLOS ONE. After careful consideration, we feel that it has merit but does not fully meet PLOS ONE’s publication criteria as it currently stands. Therefore, we invite you to submit a revised version of the manuscript that addresses the points raised during the review process.

Dear author/s,

Thank you for your submission. I have now received the reviews from the reviewers. After careful consideration, we feel that it has merit but needs revision. Please carefully address the comments, provide a line-by-line response letter, and highlight all the changes you make with different comments. If you disagree with the reviewers' comments, please write a rebuttal justifying why you disagree. Thank you

Academic Editor,Ehsan Namaziandost

We look forward to receiving your revised manuscript.

Kind regards,

Ehsan Namaziandost

Academic Editor

PLOS ONE

2. Please include a separate caption for each figure in your manuscript.

Reviewers' comments:

Reviewer's Responses to Questions

**Comments to the Author**

1. Is the manuscript technically sound, and do the data support the conclusions?

Reviewer #1: Yes

Reviewer #2: Yes

2. Has the statistical analysis been performed appropriately and rigorously? 

Reviewer #1: Yes

Reviewer #2: Yes

3. Have the authors made all data underlying the findings in their manuscript fully available?

Reviewer #1: Yes

Reviewer #2: Yes

4. Is the manuscript presented in an intelligible fashion and written in standard English?

Reviewer #1: Yes

Reviewer #2: Yes

5. Review Comments to the Author

Reviewer #1: Abstract:

Background:

Clarify the initial premise by breaking down the sentence for better readability.

Replace "have been acknowledged" with "are well-documented" for clarity.

Purpose:

Simplify the sentence structure.

Use "aims to" instead of "is to" for a more formal tone.

Ensure consistent tense usage (present tense) for a research aim.

Method:

Use "participants" rather than "students" to maintain formal research terminology.

Include an equal sign and parentheses for the mean age (mean age = 13.47 years, SD = 0.5) for clarity and precision.

Conclusion:

Use "To enhance adolescents' academic success" for a concise summary.

Ensure the recommendation flows logically from the results presented.

Introduction:

In China, test-based education is one of the most important methods of selecting talent, which puts many students under a lot of academic pressure [1]. Comment: Rephrase to: "In China, test-based education is a primary method for selecting talent, placing significant academic pressure on students [1]." This makes the sentence more concise and clear.

During this special academic climate, middle school students are facing challenges such as how to form learning habits and behaviors, deal with obstacles in study, and reduce learning pressure. Comment: Combine and streamline: "In this academic climate, middle school students face challenges in forming learning habits, overcoming study obstacles, and reducing learning pressure." This reduces redundancy.

It is evaluated and measured using a variety of criteria to determine students’ academic performance and progress, including grade assessment, academic skills and abilities, and academic outcomes [4, 5, 6]. Comment: Remove redundancy and tighten: "It is measured through various criteria, including grade assessments, academic skills, and outcomes [4, 5, 6].

To fill this gap, our study aims to comprehensively check the interplay between social support, self-efficacy, learning engagement, and academic performance. Comment: Clarify: "To fill this gap, our study aims to comprehensively examine the interplay between social support, self-efficacy, learning engagement, and academic performance."

The study’s goal is to give a comprehensive knowledge of the cumulative influence of social support, self-efficacy, and learning engagement on academic success. Comment: Simplify: "The study aims to provide a comprehensive understanding of the combined influence of social support, self-efficacy, and learning engagement on academic success."

The Social Cognitive Theory (SCT) gives a theoretical structure for developing a chain mediation model in this study. Comment: Simplify: "Social Cognitive Theory (SCT) provides the theoretical framework for developing a chain mediation model in this study.

Literature section:

The literature section seem good but some latest ref are necessary. The given paper are closely related to the nature of the paper can cited.

Zeb, A., Gan, G. G. G., Wei, O. J., & Karim, R. (2024). Examining the nexus between situational factors and job performance through the mediating role of work engagement and self‐efficacy. Journal of Public Affairs, 24(2), e2915.

Zeb, A., Goh, G. G. G., Javaid, M., Khan, M. N., Khan, A. U., & Gul, S. (2023). The interplay between supervisor support and job performance: Implications of social exchange and social learning theories. Journal of Applied Research in Higher Education, 15(2), 429-448.

Zeb, A., Ullah, R., & Karim, R. (2024). Exploring the role of ChatGPT in higher education: opportunities, challenges and ethical considerations. The International Journal of Information and Learning Technology, 41(1), 99-111.

Method:

Provide a citation for the recommended sample size standards in structural equation modeling (SEM) to support the claim of stability and reliability of the model estimate.

The sentence "The selection of these schools was based on their willingness to participate and their representation of diverse socioeconomic backgrounds." should specify how diversity in socioeconomic backgrounds was assessed and measured.

Mention the time frame during which the data collection took place.

Add a section detailing the statistical methods and software used for data analysis.

Describe how missing data was handled and any assumptions made during the analysis.

Results:

The results section seem good.

Discussion:

Mention conclusion in the one para.

Reviewer #2: 1. In the introduction, there is a lack of coherence between each paragraph.

2. In the introduction, please state the gaps in the study.

3. Please add a chapter, "literature review". 4. lines 66-98 need to be greatly condensed.

4. The lines 66~98 need to be greatly condensed and summarised in no more than one paragraph.

5. Please add sampling techniques.

6. APA formatting should not be included in the text.

7. participant characteristics should be stated in the methodology.

8 Suggest additional subheadings for research discussions.

9 Please add a section on "Conclusion".

10 The following literature is relevant to the study and is provided for reference:

The effect of online game addiction on reduced academic achievement motivation among Chinese college students: The mediating role of learning engagement

The association of short video problematic use, learning engagement, and perceived learning ineffectiveness among Chinese vocational students

Satisfaction with online study abroad predicted by motivation and self-efficacy: A perspective based on the situated expectancy–value theory during the COVID-19 epidemic

The effects of academic self-efficacy on vocational students behavioral engagement at school and at firm internships: A model of engagement-value of achievement motivation

6. PLOS authors have the option to publish the peer review history of their article (what does this mean?). If published, this will include your full peer review and any attached files.

Reviewer #1: No

Reviewer #2: No

---

## [Author Response · Author response to Decision Letter 0]

17 Jul 2024

Response to Reviewers

Title: The Chain Mediating Roles of Self-efficacy and Learning Engagement

Authors: Xiangping Zhang, Wensheng Qian

Manuscript ID: PONE-D-24-10692

Dear reviewers,

Thank you very much for your comments and professional advice. We agree with the reviewers' suggestions and will incorporate the recommended changes into the manuscript. We believe that these revisions will significantly strengthen our manuscript and provide a more robust foundation for our claims. We hope that our work can be improved again. Furthermore, we would like to show the details as follows:

Reviewer 1#

Abstract Section

Comments: In the abstract section, authors need to improve the wording, break down sentences, simplify sentence structure, switch tenses, etc. to enhance accuracy, logic, and readability.

Response: Thanks for your suggestions, you have provided us with very valuable advice to improve the quality of the abstract section. We have used your comments to improve its accuracy, logic, and readability.

1. Background was revised as “Although the effects of social support, self-efficacy, and learning engagement on academic performance are well-documented, there is still limited understanding of the specific mechanisms through which social support influences academic performance through self-efficacy and learning engagement.”

2. Purpose was revised as “This study aims to investigate the relationship between social support and adolescents' academic performance based on the Social Cognitive Theory.”

3. The method was revised as “The data was collected from 265 participants (mean age=13.47 years, SD = 0.5) in four middle schools, in the Shandong Province of China in June 2023.”

4. Conclusion was revised as “To enhance adolescents’ academic success, appropriate interventions should be implemented for teachers, parents, or other social factors, to construct a positive learning environment in and out of school to improve adolescents’ self-efficacy and learning engagement.”

Introduction Section

Comments: In the Introduction section, authors need to rephrase, combine, streamline, remove redundancy, and tighten, clarify, and simplify the sentences.

1. Rephrase “In China, test-based education is one of the most important methods of selecting talent, which puts many students under a lot of academic pressure [1].“

2. Combine and streamline: "During this special academic climate, middle school students are facing challenges such as how to form learning habits and behaviors, deal with obstacles in study, and reduce learning pressure.”

3. Remove redundancy and tighten: "It is evaluated and measured using a variety of criteria to determine students’ academic performance and progress, including grade assessment, academic skills and abilities, and academic outcomes [4, 5, 6].”

4. Clarify: "To fill this gap, our study aims to comprehensively check the interplay between social support, self-efficacy, learning engagement, and academic performance.”

5. Simplify: "The study’s goal is to give a comprehensive knowledge of the cumulative influence of social support, self-efficacy, and learning engagement on academic success.”

6. Simplify: "The Social Cognitive Theory (SCT) gives a theoretical structure for developing a chain mediation model in this study.”

Response: Thanks for your corrections, it was revised as:

1. In China, middle schools are committed to providing quality education as an essential component of compulsory education, and as such they pay special attention to the development of intellectual education. For this reason, their assessment methods focus primarily on examinations for assessing student progress, learning, and selecting talent [1, 2], which does emphasize the importance of examination results in the education process.

2. However, it cannot be ignored that this evaluation method also brings challenges to middle school students in forming learning habits, overcoming study obstacles, and reducing learning pressure [3]. 

3. It is measured through various criteria, including grade assessments, academic skills, and outcomes [13, 14].

4. To fill this gap, our research will look at the relationship between social support, self-efficacy, learning engagement, and academic performance, as well as the role of self-efficacy and learning engagement as mediators.

5. The study aims to provide a comprehensive understanding of the combined influence of social support, self-efficacy, and learning engagement on academic success with SCT.

6. The Social Cognitive Theory (SCT) proposed by Bandura in 1986, provides the theoretical framework for developing a chain mediation model in this study.

Literature Section

Comments: In the literature section, authors need to add some latest ref.

Response: As suggested by the reviewer, we have added more references to support the ideas proposed.

Method Section

Comments 1: In the Method section, authors need a citation for the recommended sample size standards in structural equation modeling (SEM) to support the claim of stability and reliability of the model estimate.

Response 1: We thank the reviewer for this helpful recommendation. We added the sample size selection criteria on Page 10 Lines 179-180:” It is explained that the proper sample size should be at least ten times the total number of variables observed due to the Structure Equation Modeling (SEM) recommendation [57].”

Comments 2: The sentence "The selection of these schools was based on their willingness to participate and their representation of diverse socioeconomic backgrounds." should specify how diversity in socioeconomic backgrounds was assessed and measured.

Response 2: Thank you for your question. The four middle schools chosen in our research were located in the urban and rural areas of Shandong Province, China. First of all, the geographical distribution of the four junior secondary schools itself reflects the diversity of China's socioeconomic structure. Secondly, factors such as the composition of students and the quality of education of four middle schools also reflect the diversity of China's socioeconomic structure. Furthermore, the diversity in socioeconomic backgrounds of the participating middle schools was measured by the place of residence of students, parental education level, and parental monthly income. In the text, we added the explanation on Page 10 Lines 185-188:” Meanwhile, these middle schools are representative of diverse socioeconomic backgrounds including a mix of urban and rural locations, varied student populations, and diverse academic performance levels, which ensure a balanced representation of both genders and different grade levels within the specified age range.”

Comments 3: Authors should mention the time frame during which the data collection took place.

Response 3: Thanks for your information. These participants were recruited from June 1 to June 30, 2023. we added it on Page 8 Lines 169-170.”

Comments 4: Add a section detailing the statistical methods and software used for data analysis.

Response 4: Thank you for your question. we added the explanation on Pages 12-13 Lines 227-233:” Data were analyzed using Amos 24.0 and SPSS 24.0. First, the Harman single-factor test was used to assess the possibility of common method bias. The sample’s characteristics were then appropriately reflected using descriptive analysis. Following that, a structural equation modeling (SEM) analysis was run to assess both the measurement and structural models. The measurement model was validated using confirmatory factor analysis, while the structural model was investigated using goodness-of-fit indices and path coefficients. Finally, the significance of mediating effects was determined using the bootstrapping method.” 

Comments 5: Describe how missing data was handled and any assumptions made during the analysis.

Response 5: Thanks for your information. we added the explanation on Page 10 Lines 179-183:” We distributed 300 survey forms, but only 265 (92.6%) completed survey forms were received for further analysis. 14 (4.7%) were rejected because of incomplete records and missing answers.”

Conclusion Section

Comments: Mention the conclusion in one paragraph.

Response: We are so grateful for this suggestion. As suggested by the reviewer, we have revised the conclusion in one paragraph as follows.

Conclusion 

The study set out to find the relationship between social support and academic performance in the context of China's test-based culture. The current data suggest that adolescents' academic performance is positively and directly impacted by social support, meanwhile, learning engagement and self-efficacy play the role of chain mediators in the association between social support and academic performance. Also, the relationship and effect of various variables are further described with the help of SCT. As evidenced by previous research studies, this article is unique and provides new insights regarding the role of social support, self-efficacy, learning engagement, and academic performance in the special academic climate of China. This research contributes to the existing academic performance literature by confirming a mediating process in the relationship among academic performance, social support, self-efficacy, and learning engagement. In the test-oriented learning climate of China, middle school students who get more help from parents, teachers, or peers can easily achieve self-efficacy and engage in learning, which in return achieve better academic performance in learning. In addition, the result of the research enhances the understanding of the relationship between social support and academic performance based on the SCT, which emphasizes the significance of social elements in determining students' self-efficacy and involvement in the learning process. Regarding the practical implications, it is conducive for educational practitioners to easier comprehend the measures of improving students' academic performance from the angle of social factors such as educators, parents, peers, etc. Regarding social support, teachers should foster an inclusive and internal classroom environment to promote respect, empathy, and cooperation in learning, such as carrying out peer mentoring programs, and cooperative learning activities. Parents should create a supportive learning environment at home, for example, building a focused atmosphere, providing dedicated study space, and minimizing external distractions. In regards to self-efficacy, teachers or parents should encourage students to participate in problem-solving activities that connect everyday learning to real-life experiences, as well as motivate adolescents to take on challenges and solve problems [80]. In addition, teachers or parents should provide timely and constructive feedback that allows students to monitor their learning progress and adjust their strategies accordingly, thereby raising students' self-efficacy. Concerning learning engagement, teachers, parents, and other social factors should work together to gain a better understanding of adolescents' needs and use tactics and skills to increase their engagement in learning through meaningful practical activities.

Reviewer 2#

Introduction Section

Comments 1: Improve the coherence between each paragraph of the Introduction.

Responsev1: We are very sorry for this kind of mistake and we have carefully reviewed the Introduction Section and the entire manuscript to improve the coherence between each paragraph.

Comments 2: State the gaps in the study.

Response 2: We are sorry to bother you with our negligence and we have added the statement of gaps in the study on Pages 4-5 Lines 63-69:” However, there is still a lack of understanding about the specific mechanisms by which social support, self-efficacy, learning engagement, and academic performance interact with one another, particularly the mediation effect of self-efficacy and learning engagement on social support and academic performance in adolescents. To fill this gap, our research will look at the relationship between social support, self-efficacy, learning engagement, and academic performance, as well as the role of self-efficacy and learning engagement as mediators.”

Comments 3: Make the literature review section a separate paragraph.

Response 3: We appreciate the reviewer for this recommendation and we have made the literature review section a separate paragraph on Pages 5-9, Line 88-160.

LITERATURE REVIEW

Social support and academic performance

Relevant research has shown that social support is a significant predictor of academic performance. It has been found that social support can significantly improve people's self-confidence [26], and motivation [27]. Researchers also have proved that parental involvement and encouragement can enhance students' focus and motivation, leading to improved academic outcomes [28, 29]. Additionally, a supportive family environment provides stability and emotional support, helping students obtain higher academic grades [30]. Social support from peers also plays a crucial role in influencing academic performance [31, 32]. For example, Wentzel noted that interactions with peers who demonstrate positive learning attitudes and behaviors can stimulate students' motivation and enhance their academic performance [33]. Importantly, it has been revealed that social support from educators has a significant positive impact on academic performance [34]. Research has indicated that individualized tutoring and additional assistance provided by teachers can address students' specific learning needs and provide effective learning guidance, thereby promoting academic progress [35]. Overall, these findings emphasize the critical role of social support in academic performance, highlighting that adolescents who get support from their parents, peers, and teachers are more inclined to achieve success in their academic pursuits. On this basis, we propose the following hypothesis.

Comments 4: The lines 66~98 need to be greatly condensed and summarised in no more than one paragraph.

Response 4: Thanks for your suggestions, we have used your comments to improve its accuracy, logic, and readability on Page 5 Lines 70-87:

“The Social Cognitive Theory (SCT), proposed by Bandura in 1986, provides the theoretical framework for developing a chain mediation model in this study. According to this theory, human behavior is influenced by three variables: personal factors (e.g., self-efficacy), behavioral factors (e.g., learning engagement and achievement), and environmental factors (e.g., social support ). Namely, individual behavior is driven and moderated by personal, behavioral, and environmental elements [23]. Past investigations have used SCT to probe into how personal cognitive or environmental factors affect academic performance among adolescents [24, 25], with less attention paid to the interplay between social support, self-efficacy, learning engagement, and academic performance within the framework of SCT. Based on the above, the study aims to provide a comprehensive understanding of the combined influence of these four factors with SCT. 

This study takes a broader approach, examining the interrelationships and mediating effects of social support, self-efficacy, learning engagement, and academic performance. As a result, this study differs from previous research, which focused on the impact of single factors on academic performance. The significance of this study stems from its contribution to bridging the existing research gap and providing a better understanding of the factors that influence students' academic success. By investigating the interactive effects and mediating roles of these factors, the study proposes to offer valuable insights into the complex dynamics that influence academic performance among adolescents.”

Method Section

Comments 1: Please add sampling techniques.

Response 1: We appreciate the reviewer for this kind of recommendation and we have added the sampling techniques on Page 9-10 Line 168-178:” The present study employed the simple random sampling method to recruit Chinese middle school students to voluntarily take part in the questionnaire collection. The data was collected from 1 June to 30 

---

## [Decision Letter · Decision Letter 1]

25 Jul 2024

PONE-D-24-10692R1The effect of social support on academic performance among adolescents: The chain mediating roles of self-efficacy and learning engagementPLOS ONE

Dear Dr. Qian,

Thank you for submitting your manuscript to PLOS ONE. After careful consideration, we feel that it has merit but does not fully meet PLOS ONE’s publication criteria as it currently stands. Therefore, we invite you to submit a revised version of the manuscript that addresses the points raised during the review process.

We look forward to receiving your revised manuscript.

Kind regards,

Ehsan Namaziandost

Academic Editor

PLOS ONE

Additional Editor Comments:

Dear author/s,

Thank you for your submission. I have now received the reviews from the reviewers. After careful consideration, we feel that it has merit but needs revision. Please carefully address the comments, provide a line-by-line response letter, and highlight all the changes you make with different comments. If you disagree with the reviewers' comments, please write a rebuttal justifying why you disagree. Thank you

Academic Editor,

Ehsan Namaziandost

Reviewers' comments:

Reviewer's Responses to Questions

**Comments to the Author**

1. If the authors have adequately addressed your comments raised in a previous round of review and you feel that this manuscript is now acceptable for publication, you may indicate that here to bypass the “Comments to the Author” section, enter your conflict of interest statement in the “Confidential to Editor” section, and submit your "Accept" recommendation.

Reviewer #1: (No Response)

2. Is the manuscript technically sound, and do the data support the conclusions?

Reviewer #1: Partly

3. Has the statistical analysis been performed appropriately and rigorously? 

Reviewer #1: Yes

4. Have the authors made all data underlying the findings in their manuscript fully available?

Reviewer #1: Yes

5. Is the manuscript presented in an intelligible fashion and written in standard English?

Reviewer #1: No

6. Review Comments to the Author

Reviewer #1: Abstract: follow the given format in the abstract like purpose, method, results, findings, practical implications, and originality/value.

Introduction: lines 20-21 the sentence is clear but could be made more concise. Lines 24-26 The phrase "it cannot be ignored that" can be simplified. Lines 27-29 Consider rephrasing for better flow. Lines 29-30 the sentence is lengthy and could be broken into two for clarity. Lines 34-36 rephrase for clarity and remove redundancy. Lines 45-47 Clarify the importance of self-efficacy and learning engagement. Lines 54-59 Combine and streamline for clarity. lines 63-66 Streamline and emphasize the research gap.

Literature: The following paper can be cited its closely link with nature of the given paper.

Zeb, A., Goh, G. G. G., Javaid, M., Khan, M. N., Khan, A. U., & Gul, S. (2023). The interplay between supervisor support and job performance: Implications of social exchange and social learning theories. Journal of Applied Research in Higher Education, 15(2), 429-448.

Rehman, F. U., Ismail, H., Al Ghazali, B. M., Asad, M. M., Shahbaz, M. S., & Zeb, A. (2021). Knowledge management process, knowledge based innovation: Does academic researcher’s productivity mediate during the pandemic of covid-19?. Plos one, 16(12), e0261573. Zeb, A., Ullah, R., & Karim, R. (2024). Exploring the role of ChatGPT in higher education: opportunities, challenges and ethical considerations. The International Journal of Information and Learning Technology, 41(1), 99-111. Rehman, F. U., & Zeb, A. (2023). Investigating the nexus between authentic leadership, employees’ green creativity, and psychological environment: evidence from emerging economy. Environmental Science and Pollution Research, 30(49), 107746-107758.

Method: seem good.

Results: If HTMT approach is possible in AMOS can be mentioned to ensure the discriminant validity issue.

Discussion: Two headings one is Practical implication and another theoretical implications can be mentioned.

7. PLOS authors have the option to publish the peer review history of their article (what does this mean?). If published, this will include your full peer review and any attached files.

Reviewer #1: No

---

## [Author Response · Author response to Decision Letter 1]

4 Sep 2024

Title: The effect of social support on academic performance among adolescents: The chain mediating roles of self-efficacy and learning engagement 

Authors: Xiangping Zhang, Wensheng Qian

Manuscript ID: PONE-D-24-10692

Dear reviewer,

Thank you for your letter and for reviewer’ comments concerning our manuscript entitled “The effect of social support on academic performance among adolescents: The chain mediating roles of self-efficacy and learning engagement”. All of those remarks are relevant and beneficial for the purpose of rewriting and enhancing our paper. We have thoroughly examined every comments and have made diligent corrections. We are confident that these adjustments will greatly enhance our manuscript and establish a strong basis for our assertions. We aspire for continued enhancement of our work.The responses to the reviewer's comments are as follows:

Reviewer 1#

Abstract Section

Comments: Authors must adhere to the prescribed structure for the abstract, which includes sections on the purpose, method, results, findings, practical implications, and originality/value.

Response: We appreciate your ideas, since they have given us great guidance on enhancing the quality of the abstract part. We have made revisions to this section as follows:

Purpose- While the impact of social support on academic performance is acknowledged, the specific mechanisms by which social support affects academic performance, particularly through self-efficacy and learning engagement, remain poorly understood. This study aims to examine the correlation between social support and academic achievement among Chinese middle school students, framed within the Social Cognitive Theory. It also seeks to explore the mediating roles of self-efficacy and learning engagement in this relationship. 

Method- Data was collected from 265 individuals (mean age = 13.47 years, SD = 0.5) in four middle schools in Shandong Province, China in June 2023, using the simple random sample technique. Participants completed the questionnaires independently, and the data was analyzed using the structural equation model (SEM) in AMOS 24.0 and SPSS 24.0. 

Results- Social support and academic performance have a direct and significant relationship with the SCT among middle school students. In addition, social support indirectly and positively affects academic performance through self-efficacy and learning engagement. The results also highlight self-efficacy as a key factor linking social support with academic performance.

Practical implications- This study offers valuable insights into the role of social support in Chinese middle school students’ academic achievement, particularly by examining the impact of self-efficacy and learning engagement. These valuable findings may guide policymakers in creating a supportive educational environment both inside and outside the classroom, enhancing adolescents’ self-confidence and engagement in learning. 

Originality- This study contributes to the theoretical understanding of social support by investigating the mechanisms through which it impacts academic achievement. It clarifies the complex interactions among social support, self-efficacy, learning engagement, and academic achievement, with particular emphasis on the mediating roles of self-efficacy and learning engagement within the Chinese context.

Introduction Section

Comments: Revise the Introduction section to improve linguistic conciseness, streamline the information, and ensure clarity.Introduction: lines 20-21 the sentence is clear but could be made more concise. Lines 24-26 The phrase "it cannot be ignored that" can be simplified. Lines 27-29 Consider rephrasing for better flow. Lines 29-30 the sentence is lengthy and could be broken into two for clarity. Lines 34-36 rephrase for clarity and remove redundancy. Lines 45-47 Clarify the importance of self-efficacy and learning engagement. Lines 54-59 Combine and streamline for clarity. lines 63-66 Streamline and emphasize the research gap.

Response: Thanks for your corrections, it was revised as:

Lines 20-21 are revised as: Within China's compulsory education system, middle school serves as a crucial intermediary, bridging different levels of education. At this stage, academic performance is a significant indicator of students' information acquisition and their potential for future studies. It is used to evaluate student progress, learning, and talent selection [1, 2].

Lines 24-26 are revised as: In China, academic success is commonly assessed through exam scores in subjects such as Chinese, Math, and English [3, 4]. Nevertheless, this assessment approach presents challenges for middle school students in developing study routines, overcoming academic difficulties, and managing time and anxiety [5]. 

Lines 27-29 are revised as: Learning is predominantly a cognitive endeavor that involves the acquisition of knowledge through various social activities [6]. Chinese students, however, have limited opportunities for social engagement, primarily confined to interactions within the classroom, on campus, and within their families. 

Lines 29-30 are revised as: Social support refers to the resources obtained through social interactions, which reflect the degree of connection between an individual and their community.

Lines 34-36 are revised as: It serves as a defensive shield against negative emotions and stress [7, 8], providing a sense of being valued and supported by others when needed [9].

Lines 45-47 are revised as: These two psychological constructs are closely related to academic performance [20]. Self-efficacy acts a crucial role in motivating individuals to achieve their goals, encouraging them to take risks, and reaching their academic outcomes [21,22]. 

Lines 54-59 are revised as:Learning engagement refers to the active participation of students in the educational process, which positively impacts on academic performance. Engaged students are typically more motivated, committed, and willing to invest the effort needed to participate in discussions, solve problems, and achieve academic success [25, 26]. 

Lines 63-66 are revised as: However, little attention has been paid to the influence of social support on the academic performance of Chinese middle school students. Furthermore, this study identifies a gap in understanding the precise mechanisms by which social support affects academic performance through self-efficacy and learning engagement, as outlined in relevant theoretical frameworks.

Literature Section

Comments: In the literature section, authors need to add some latest ref.

Response: As suggested by the reviewer, we have added more references to support the ideas proposed.

2.Zeb A, Ullah R, Karim R. Exploring the role of ChatGPT in higher education: opportunities, challenges and ethical considerations. The International Journal of Information and Learning Technology. 2024; 41(1):99-111. https://doi.org/10.1108/ijilt-04-2023-0046

6.Rehman FU, Ismail H, Al Ghazali B M, Asad MM, Shahbaz MS, Zeb A. Knowledge management process, knowledge based innovation: Does academic researcher’s productivity mediate during the pandemic of covid-19?. Plos one. 2021;16(12):e0261573. https://doi.org/10.1371/journal.pone.0261573

22.Rehman FU, Zeb A. Investigating the nexus between authentic leadership, employees’ green creativity, and psychological environment: evidence from emerging economy. Environmental Science and Pollution Research. 2023; 30(49):107746-107758. https://doi.org/10.1007/s11356-023-29928-1

30.Zeb A, Gan GGG, Javaid M, Khan MN, Khan AU, Gul S. The interplay between supervisor support and job performance: Implications of social exchange and social learning theories. Journal of Applied Research in Higher Education. 2023; 15(2):429-448. https://doi.org/10.1108/jarhe-04-2021-0143

42.Zeb A, Gan GGG, Wei OJ, Karim R. Examining the nexus between situational factors and job performance through the mediating role of work engagement and self‐efficacy. Journal of Public Affairs. 2024; 24(2):e2915. https://doi.org/10.1002/pa.2915

Results Section

Comments:If the HTMT technique is applicable in AMOS, it might be mentioned to address the issue of discriminant validity.

Response:In response to the reviewer's suggestion, we have implemented the HTMT technique to resolve the problem of discriminant validity.

To validate discriminant validity, this study employed the Heterotrait-Monotrait (HTMT) criterion. According to Kline [68], and Henseler et al. [69], an acceptable HTMT value should remain below 0.85. Table 4 indicates that discriminant validity was achieved. 

Table 4. Heterotrait-Monotrait Ration (HTMT)

Potential variable Social support Self-efficacy Learning engagement

Social support 

Self-efficacy 0.403 

Learning engagement 0.508 0.326 

Discussion Section

Comments: Two headings, "Practical Implications" and "Theoretical Implications," can be included in Discussion section.

Response: We are so grateful for this suggestion. As suggested by the reviewer, we have revised the discussion as follows.

Theoretical implication 

This study contributes to the existing theoretical knowledge by underscoring the influence of social support on the academic performance of middle school students within the framework of Social Cognitive Theory. It supports the notion that social support plays a significant role in academic achievement by elucidating the complex interactions among social support, self-efficacy, learning engagement, and academic performance. Furthermore, this research builds on previous empirical studies that have established a link between social support and academic performance. By confirming these findings within the context of the Chinese compulsory education system and emphasizing the mediating roles of self-efficacy and learning engagement, this study enhances the theoretical understanding of the relationship between social support and academic achievement among middle school students in China.

Practical implication

Regarding the practical implications, it is essential for educational practitioners to understand how to enhance students' academic achievement by considering social factors such as the roles of teachers, parents, and peers. To strengthen social support, teachers should create an inclusive and cohesive classroom environment that fosters respect, understanding, and collaboration among students. This can be accomplished through initiatives like peer mentorship programs and collaborative learning activities. Parents also play a vital role in establishing a conducive learning environment at home. They can do this by promoting a focused atmosphere, designating a dedicated study area, and minimizing external distractions. To enhance self-efficacy, it is important for both teachers and parents to encourage students to participate in problem-solving activities that relate to real-life situations. Additionally, they should motivate adolescents to embrace challenges and seek solutions, thereby helping them develop confidence in their abilities [86]. Furthermore, it is essential for educators and guardians to provide timely and constructive feedback that allows students to monitor their learning progress and adjust their approaches accordingly. This kind of feedback can significantly enhance students' self-efficacy and belief in their own abilities. In terms of learning engagement, it is important for teachers, parents, and other social variables to collaborate in order to develop a comprehensive understanding of adolescents' needs. By employing effective strategies and techniques, they can foster greater involvement in learning through meaningful and practical activities. This coordinated effort will not only engage students more deeply but also support their overall academic development.

We deeply value the time and effort dedicated by the reviewers in assessing our article. We eagerly anticipate any further feedback or recommendations.

---

## [Decision Letter · Decision Letter 2]

23 Sep 2024

The effect of social support on academic performance among adolescents: The chain mediating roles of self-efficacy and learning engagement

PONE-D-24-10692R2

Dear Dr. Qian,

We’re pleased to inform you that your manuscript has been judged scientifically suitable for publication and will be formally accepted for publication once it meets all outstanding technical requirements.

Kind regards,

Ehsan Namaziandost

Academic Editor

PLOS ONE

Additional Editor Comments (optional):

Reviewers' comments:

Reviewer's Responses to Questions

**Comments to the Author**

1. If the authors have adequately addressed your comments raised in a previous round of review and you feel that this manuscript is now acceptable for publication, you may indicate that here to bypass the “Comments to the Author” section, enter your conflict of interest statement in the “Confidential to Editor” section, and submit your "Accept" recommendation.

Reviewer #1: All comments have been addressed

2. Is the manuscript technically sound, and do the data support the conclusions?

Reviewer #1: Yes

3. Has the statistical analysis been performed appropriately and rigorously? 

Reviewer #1: Yes

4. Have the authors made all data underlying the findings in their manuscript fully available?

Reviewer #1: Yes

5. Is the manuscript presented in an intelligible fashion and written in standard English?

Reviewer #1: Yes

6. Review Comments to the Author

Reviewer #1: The coefficient of determination value R square. I not found find if not available. Please include it.

7. PLOS authors have the option to publish the peer review history of their article (what does this mean?). If published, this will include your full peer review and any attached files.

Reviewer #1: No

---

## [Editor Report · Acceptance letter]

11 Dec 2024

PONE-D-24-10692R2 

PLOS ONE

Dear Dr. Qian, 

I'm pleased to inform you that your manuscript has been deemed suitable for publication in PLOS ONE. Congratulations! Your manuscript is now being handed over to our production team.

Kind regards, 

on behalf of

Dr. Ehsan Namaziandost 

Academic Editor

PLOS ONE